# *Candida haemulonii* complex, an emerging threat from tropical regions?

Ugo Françoise[1¤a], Marie Desnos-Ollivier[2], Yohann Le Govic[1¤b], Karine Sitbon[2], Ruddy Valentino[3], Sandrine Peugny[4], Taieb Chouaki[5], Edith Mazars[6], André Paugam[7], Muriel Nicolas[8], Nicole Desbois-Nogard[1☯], Olivier Lortholary[2,9,10☯]*, French Mycoses Study Group[¶]

**1** Laboratoire de Parasitologie-Mycologie, Centre Hospitalier Universitaire de Martinique, Fort-de-France, Martinique, France, **2** Centre National de Référence des Mycoses invasives et Antifongiques, Département de Mycologie, Institut Pasteur, Université de Paris Cité, Paris, France, **3** Service de réanimation, Centre Hospitalier Universitaire de Martinique, Fort-de-France, Martinique, France, **4** Unité de Maladies Infectieuses et Tropicales, Centre Hospitalier de Cayenne, Cayenne, Guyane Française, France, **5** Laboratoire de Parasitologie-Mycologie, Centre Hospitalier Universitaire Amiens-Picardie, Amiens, France, **6** Laboratoire de Parasitologie-Mycologie, Centre Hospitalier de Valenciennes, Valenciennes, France, **7** Laboratoire de Parasitologie-Mycologie, Centre Hospitalier Universitaire Cochin, Assistance Publique-Hôpitaux de Paris, Paris, France, **8** Laboratoire de Parasitologie-Mycologie, Centre Hospitalier Universitaire de Guadeloupe, Pointe-à-Pitre, Guadeloupe, France, **9** Institut Imagine, Paris, France, **10** Hôpital Universitaire Necker-Enfants Malades, Assistance Publique-Hôpitaux de Paris, Paris, France

☯ These authors contributed equally to this work.
¤a Current address: Unité de Maladies Infectieuses et Tropicales, Centre Hospitalier de Cayenne, Cayenne, Guyane Française, France
¤b Current address: Laboratoire de Parasitologie-Mycologie, Centre Hospitalier Universitaire Amiens-Picardie, Amiens, France
¶ Membership of French Mycoses Study Group is provided in Supporting Information file S1 Acknowledgments.
* olivier.lortholary@wanadoo.fr

**Data Availability Statement:** For data from the RESSIF and YEASTS networks The data used for this publication are considered as pseudonymized personal data. As such the GDPR as well as the

## Abstract

### Background

*Candida haemulonii* complex-related species are pathogenic yeasts closely related to *Candida auris* with intrinsic antifungal resistance, but few epidemiological data are available.

### Methodology/Principal findings

We analyzed clinical and demographic characteristics of patients with fungemia due to *C. haemulonii* complex and related species (*C. pseudohaemulonii*, *C. vulturna*) reported in France during 2002–2021, and compared them to data of *C. parapsilosis* fungemia, as they all can be commensal of the skin. We also conducted a study on adult inpatients and outpatients colonized by *C. haemulonii* complex, managed at the University Hospital of Martinique during 2014–2020. Finally, we performed a literature review of fungemia due to *C. haemulonii* complex and related species reported in Medline (1962–2022).

In total, we identified 28 fungemia due to *C. haemulonii* complex in France. These episodes were frequently associated with bacterial infection (38%) and high mortality rate (44%), and differed from *C. parapsilosis* fungemia by their tropical origin, mainly from Caribbean and Latin America. All isolates showed decreased *in vitro* susceptibility to

French Law pertaining data protection apply to the use of this dataset. For data access request, a requester can contact: Pr Fanny Lanternier, head of NRCMA, cnrma@pasteur.fr The requester shall be informed that access to and reuse of this dataset will require the obtention of (i) an authorization from the French Supervisory Authority regarding data protection (the CNIL) and (ii) an ethics opinion from the competent French ethics committee (the CESREES) For data from the University Hospital of Martinique Access to the data of the University Hospital of Martinique can be requested from the institution's Data Protection Officer (DPO): dpo@chu-martinique.fr.

**Funding:** This work was supported by Santé Publique France (UF, MDO,YLG, KS, RV, SP, TC, EM, AP, MN,NDG,OL) and Institut Pasteur (UF, MDO,YLG, KS, RV, SP, TC, EM, AP, MN,NDG, OL). The funders had no role in study design, data collection, analysis, or interpretation of data.

**Competing interests:** The authors have declared that no competing interests exist.

amphotericin B and fluconazole. In Martinique, we found that skin colonization was frequent in the community population, while colonization was strongly associated with the presence of foreign devices in ICU patients. The literature review identified 274 fungemia episodes, of which 56 were individually described. As in our national series, published cases originated mainly from tropical regions and exhibited high crude mortality.

## Conclusions/Significance

Multidrug-resistant *C. haemulonii* complex-related species are responsible for fungemia and colonization in community and hospital settings, especially in tropical regions, warranting closer epidemiological surveillance to prevent a potential *C. auris*-like threat.

### Author summary

Yeasts of the *C. haemulonii* complex (*C. haemulonii sensu stricto*, *C. duobushaemulonii*, *C. haemulonii var. vulnera*) and related species (*C. pseudohaemulonii* and *C. vulturna)*, are phylogenetically close to *Candida auris*, but their distribution, pathophysiology and antifungal resistance profiles are poorly known. This work provides the first epidemiological data on these yeasts in France. Fungemia caused by these yeasts were mainly identified in tropical overseas regions (French West Indies, French Guiana), and occurred in patients with risk factors for candidemia, particularly a cutaneous portal of entry. This work highlights the skin carriage of these yeasts in these tropical regions, and their ability to colonize foreign devices. Like for *Candida parapsilosis*, catheters are the main pathway for fungemia but mortality seems higher with yeasts of the *C. haemulonii* complex and related. They commonly show *in vitro* resistance to many antifungal agents, notably fluconazole and amphotericin B, which are still frequently used as first and second-line treatments for candidemia worldwide. The findings from the literature review are consistent with these overall results. These observations justify closer epidemiological surveillance in the concerned regions to prevent a potential *C. auris*-like threat.

## Introduction

Since the 2010s [1], multidrug-resistant *Candida auris* [2], with its ability to survive on prosthetic materials and spread among patients, has become a spreading healthcare-associated fungus [3–5]. Within the *Metschnikowiaceae* clade, yeasts of the *Candida haemulonii* complex (*C. haemulonii sensu stricto*, *C. haemulonii var. vulnera*, and *C. duobushaemulonii*), *C. pseudohaemulonii*, and *C. vulturna* are phylogenetically closely related to *C. auris* and share several pathogenicity-related traits, like adhesion on prosthetic materials [6,7] phenotypic switching [8], and multidrug resistance [7,9–11]. In fact, misidentifications by biochemical methods are frequent, even with updated databases [12]. Few cases of *C. haemulonii* complex infection have been reported since its description [13,14], but since the 2000s, several fungemia have been reported, especially in tropical areas [10,15–17]. These yeasts present often high MICs for fluconazole [18], the first-line antifungal for treating fungemia in many low-income countries, and exhibit a decreased susceptibility to amphotericin B, often used as salvage therapy. Isolates of this complex are usually susceptible to echinocandins, although echinocandin-resistant strains have been isolated from human samples [19].

In France, rare yeasts are responsible for nearly 7.4% of fungemia [20] but there are no major epidemiological studies focused on this complex. Consequently, risk factors and clinical outcome of fungemia due to *Candida haemulonii* complex-related species remain largely unknown [21].

In this context, we carried out a study in mainland France and French overseas territories including (i) epidemiology of fungemia caused by *C. haemulonii* complex, *C. pseudohaemulonii* and *C. vulturna* (hereinafter referred to as "fungemia series"), (ii) a case-control study comparing these fungemia to those due to *C. parapsilosis*, a skin commensal, and (iii) a study of colonization in outpatients and inpatients admitted to intensive care units (ICU) in Martinique. Finally, we conducted a literature review on fungemia caused by these species during the past six decades.

## Methods

### Ethics statement

The fungemia series was carried out in compliance with French law and the declaration of Helsinki (as adopted in 2000) and was approved by the French Mycoses Study Group scientific council. The surveillance of the NRCMA was approved by the Institut Pasteur Institutional Review Board #1 (#2009–34/IRB) and the "Commission Nationale de l'Informatique et des Libertés" according to the French regulation. The study in Martinique has been approved by the IRB of the Martinique university hospital (#2020/064).

### Epidemiology of fungemia in France

We collected clinical data related to fungemia caused by *C. haemulonii* complex, *C. pseudohaemulonii* and *C. vulturna* from two French databases, from their creation until 31st July 2021. Both programs were launched by the French National Reference Centre for Invasive Mycoses & Antifungals (NRCMA): the YEASTS surveillance program, initiated in 2002 with the participation of 27 university or cancer hospitals in the Paris area [22] and the French Surveillance Network of Invasive Fungal Infections (RESSIF) program, initiated in 2012 with the participation of 29 hospitals from mainland France and overseas [20].

### Case-control study

The above-described series constitutes the case population. We chose fungemia caused by *C. parapsilosis* from the RESSIF database as the control population, rather than *C. albicans* because preliminary information revealed the presence of these study species on human skin.

### Colonization in Martinique

We enrolled patients with sample containing *C. haemulonii* complex yeasts, considered to be non-pathogenic (i.e. not being responsible for infection), at the Martinique University hospital, between April 2014 and August 2020. We retained cases involving adult patients with putative community-acquired colonization (referred to as "outpatients", including those hospitalized <48h) or hospitalized in ICU. Lack of pathogenicity was retained by the clinician according to clinical context and source of sampling.

### Literature review

We performed a literature review on Medline database, from January 1st, 1962 to September 1st, 2022 with the following search terms in all fields: "*Candida haemulonii*" OR "*Torulopsis haemulonii*" OR "*Candida duobushaemulonii*" OR "*Candida pseudohaemulonii*" OR "*Candida*

*vulturna*". We included all reported cases of *C. haemulonii* complex, *C. pseudohaemulonii*, and *C. vulturna* fungemia identified by sequencing, for which individual clinical information and/ or susceptibility to antifungals were available.

## Isolate characterization

**Fungemia series.**   All bloodstream isolates of *C. haemulonii* complex and related species were identified at the NRCMA by sequencing of ITS1–5.8S–ITS2 and D1/D2 regions of the ribosomal DNA, using V9D/LS266 and NL1/NL4 primers as previously described [23]. Antifungal susceptibility testing to micafungin, amphotericin B, and fluconazole, was performed using the broth microdilution method published by EUCAST (v 7.3.2 valid from 22 April, 2020), with a modification for micafungin medium as previously described [24].

**Colonization in Martinique.**   From 2014 to 2017, yeasts were identified by culture on chromogenic media chromID Candida agar (BioMerieux, Marcy l'Étoile, France) combined with carbon assimilation testing using the API ID32C system (BioMerieux, Marcy l'Étoile, France). As no API code matches the complex, strains identification was made by matching profiles with those of an internal database of isolates previously identified by sequencing. Furthermore, carbon assimilation identification does not distinguish species within the complex, isolates were identified as *C. haemulonii* complex-related yeasts.

Since 2017, species identification has been performed routinely by matrix-assisted laser desorption/ionization time of flight mass spectroscopy (MALDI-TOF MS) using the Biotyper system (Bruker Daltonics, Billerica, MA, USA). Nine strains isolated before 2017 were rechecked, showing no discrepancy with biochemical analysis (**S1 Table**). Nonetheless, since the distinction between *C. haemulonii sensu stricto* and *C. haemulonii var. vulnera* is currently not possible without sequencing [25,26], such isolates only identified by MALDI-TOF MS were considered as *C. haemulonii sensu lato* or *C. duobushaemulonii*.

## Statistical analysis

Analyses were performed on isolates from single episodes of fungemia, each isolate corresponding to a single patient. Associated factors were studied according to the species. We selected the variables significantly associated with any of the two groups at the threshold p<0.20. We developed an explanatory model using logistic regression, with a manual stepwise procedure guided by the Akaike Information Criterion. Explanatory model results were reported as adjusted odds ratios (aOR) with 95% confidence intervals (95%CI). Data were analyzed using R software (version 4.0.2). The map was created from a fla-shop.com background map (https://www.fla-shop.com/svg/, CC BY 4.0 license) modified with inkscape software.

## Results

### Fungemia series

Between October 1st, 2002, and July 31st, 2021, among the 12 032 candidemia episodes reported in YEASTS and RESSIF, 28 (0.23%) were caused by *C. haemulonii* complex, *C. pseudohaemulonii*, or *C. vulturna* (**Table 1**). The incidence appears to increase over time (**S1 Fig**), with 15/28 (53.6%) cases reported between 2017 and 2021. Notifications from the French West Indies and French Guiana accounted for 71.4% (20/28) of all cases.

Subjects were mostly men (16, 57.1%) and the median age was 67 years (Interquartile Range (IQR) = 46–72); 81.4% of them lived or had stayed in tropical regions. Twenty-six patients (92.9%) had at least one of the following comorbidities: solid cancer, hematological

**Table 1. Characteristics of patients with fungemia due to *C. haemulonii* complex, *C. pseudohaemulonii* and *C. vulturna* in France between 2002 and 2021 (YEASTS and RESSIF programs), and *C. parapsilosis* in France between 2012 and 2021 (RESSIF).**

| Characteristics | *C. haemulonii* complex-related species | *C. parapsilosis* |
|---|---:|---:|
| Total | 28 | 942 |
| Hospital (%) | | |
| Martinique university hospital (FWI) | **10 (35.7)** | **62 (6.6)** |
| Cayenne hospital (French Guiana) | **9 (32.1)** | **6 (0.6)** |
| Guadeloupe university hospital (FWI) | 1 (3.6) | 1 (0.1) |
| Paris aera hospitals | 5 (17.9) | 200 (21.2) |
| Other hospitals in mainland France | **3 (10.7)** | **673 (71.4)** |
| Birth continent (%) | | |
| Africa | 1/20 (5.0) | 35/456 (7.6) |
| America | **18/20 (90.0)** | **62/456 (13.5)** |
| Europe | **1/20 (5.0)** | **359/456 (78.2)** |
| Travels during the previous year (%) | | |
| Africa | **2/22 (9.1)** | **14/216 (6.5)** |
| Latin America | **2/22 (9.1)** | **1/216 (0.5)** |
| West Indies | 4/22 (18.2) | 48/216 (22.2) |
| None | 14/22 (63.6) | 148/216 (68.5) |
| Median age in years [IQR] | 67 [46–72] | 62 [46–72] |
| Male (%) | 16 (57.1) | 625 (66.3) |
| Solid cancer (%) | 8/24 (33.3) | 281 (29.8) |
| Hematological malignancy (%) | **5/26 (19.2)** | **118 (12.5)** |
| Chronic kidney failure (%) | 3/24 (12.5) | 102 (10.8) |
| Liver cirrhosis (%) | **2/24 (8.3)** | **43 (4.6)** |
| Diabetes mellitus (%) | 3/24 (12.5) | 129 (13.7) |
| HIV (%) | **2/23 (8.7)** | **13/412 (3.2)** |
| Corticosteroids[a] (%) | 2 (7.1) | 85 (9.0) |
| Other immunosuppressive treatment (%) | 4 (14.3) | 200 (21.2) |
| Surgery during the previous month (%) | 10/26 (38.5) | 358 (38.0) |
| Exposure to antifungal agent in the previous month (%) | | |
| Amphotericin B | 1/24 (4.2) | 11/916 (1.2) |
| Echinocandin | **0 (0.0)** | **88/916 (9.6)** |
| Fluconazole | 1/24 (4.2) | 9/916 (1.0) |
| Voriconazole | 2/24 (8.3) | 7/916 (0.8) |
| Posaconazole | 0 (0.0) | 1/916 (0.1) |
| Isavuconazole | 0 (0.0) | 5/916 (0.5) |
| None | 20/24 (83.3) | 804/916 (87.7) |
| Central venous catheter (%) | **15/22 (68.2)** | **693/753 (92.6)** |
| Context of bacterial infection (%) | **11/24 (45.8)** | **180 (19.1)** |
| Intensive Care Unit (%) | **10/26 (38.5)** | **315 (33.4)** |
| Shock (%) | **3/24 (12.5)** | **52 (5.5)** |
| Antifungal treatment (%) | | |
| Amphotericin B | 0/26 (0.0) | 46/922 (5.0) |
| Echinocandin | 15/26 (58.7) | 472/922 (51.2) |
| Fluconazole | 3/26 (11.5) | 313/922 (33.9) |
| Itraconazole | 1/26 (3.8) | 0/922 (0.0) |
| Voriconazole | 0/26 (0.0) | 16/922 (1.7) |
| Posaconazole | 0/26 (0.0) | 3/922 (0.3) |

*(Continued)*

**Table 1.** (Continued)

| Characteristics | *C. haemulonii* complex-related species | *C. parapsilosis* |
|---|---|---|
| None | 7/26 (26.9) | 86/922 (9.3) |
| All-cause mortality at 3 months (%) | **11/25 (44.0)** | **192/757 (25.4)** |
| Median time to death in days [IQR] | 18 [7–41] | 8 [2–18] |

FWI: French West Indies, IQR: Interquartile Range

[a]≥ 0.3mg/kg for ≥ 1 month

For each case, demographics, underlying conditions, initial antifungal treatment, and 3-month survival status were collected using a standardized questionnaire through a secure website.

The date of the fungemia corresponded to the date of the first positive blood culture.

Surgery and antifungal pre-exposure were considered in the 30 days prior to fungemia.

malignancy, recent surgical procedure, central venous catheter, context of bacterial infection, hospitalization in ICU; Nine of them (32.1%) presented at least three of these comorbidities.

Four distinct species were identified. *Candida haemulonii sensu stricto* was the dominant species (16, 57.1%), followed by *C. duobushaemulonii* (8, 28.6%) and *C. vulturna* (3, 10.7%). *C. pseudohaemulonii* was identified once (1, 3.6%), whereas no *C. haemulonii var. vulnera* was retrieved. The distribution of species varied according to age, with a median age of 67 and 70 years for *C. haemulonii* and *C. duobushaemulonii*, respectively, compared with 46 and 27 years for *C. pseudohaemulonii* and *C. vulturna*, respectively.

Among the three patients who presented fungemia due to *C. vulturna*, two had a history of cancer and were pre-exposed to voriconazole, and 2 had recent surgery. The only *C. pseudohaemulonii* episode occurred in a 46-year-old woman from Brazil who was hospitalized in French Guiana for gastric linitis. She had no bacterial co-infection or recent surgery.

All-cause mortality at 3 months after diagnosis was 44% (11/25), of which 7 (63%) occurred in the first 30 days and 5 (45%) in the first 15 days. Mortality increased with age, from 3 (27.2%) in the <67 age group to 57.1% in the ≥67 age group. Two patients survived despite the absence of antifungal treatment. They were under 30 years old, had no underlying disease, and benefited from prompt catheter replacement.

Antifungal susceptibility testing could be performed on 23/28 strains. The median MIC values for amphotericin B are high, especially for *C. duobushaemulonii* (2 mg/l). Oppositely, those for fluconazole were lower for *C. duobushaemulonii*, yet still high (16 mg/l). The median MIC values for micafungin were ≤0.250 mg/l for all strains (**Fig 1**).

## Case-control study

Between 2012 and 2021, 942 episodes of *C. parapsilosis* fungemia in adults were declared within the RESSIF database (**Table 1**). The geographical location of the patient in overseas territories was significantly associated with the development of *C. haemulonii* complex-related species fungemia (aOR = 70.77, 95%CI = 23.67–280.01), followed by the context of bacterial infection (aOR = 2.91, 95%CI = 1.10–7.64) (**Fig 2A**).

Three-month all-cause mortality was mainly associated with intensive care management and underlying comorbidities (**Fig 2B**). After adjusting for patient comorbidities, presence of bacterial infections, and level of care, fatal outcomes tended to be more frequent in *C. haemulonii* complex-related species fungemia than in *C. parapsilosis* fungemia (aOR = 2.39, 95% CI = 0.95–5.86).

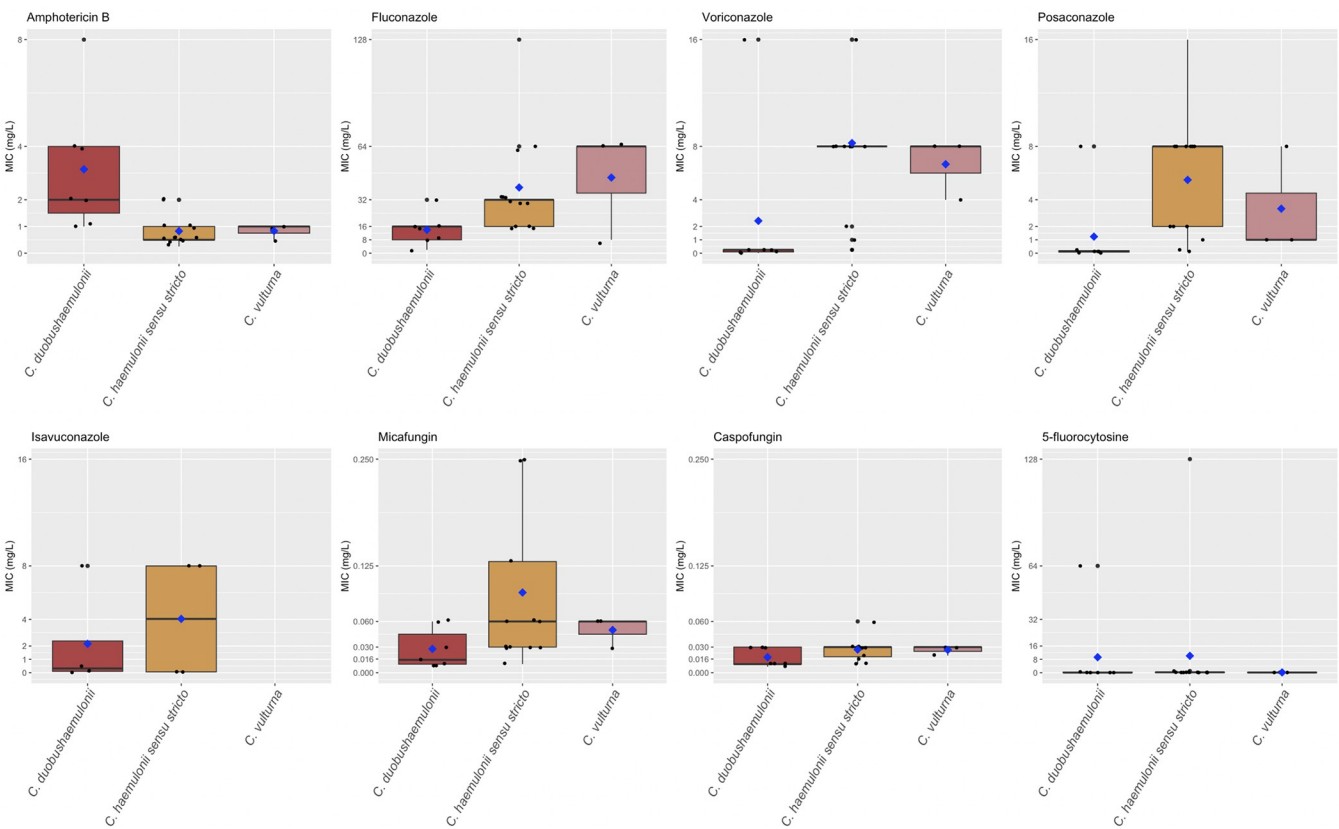

**Fig 1. Susceptibility profiles to antifungals of the strains of the French fungemia series.** By EUCAST broth micro dilution reference method. *C. duobushaemulonii* = 7, *C. haemulonii sensu stricto* = 13, *C. vulturna* = 3. Bold black bar: median MIC value; Blue diamond: mean MIC value.

## Colonization in Martinique

We identified 119 *C. haemulonii* complex colonizations, in 116 adults with consultable medical records (**S2 Fig**). Of the 87 colonizations in outpatients (**Table 2**), 61 (70.1%) were cutaneous and 12 (13.8%) were ophthalmologic.

Outpatients with skin colonization were predominantly women (41/61, 67.2%); the mean age was 61.3 years; all lived in Martinique and 24.6% (15/61) were unemployed. Their main comorbidities were diabetes mellitus (12, 19.7%), solid cancer (12, 19.7%), and immune deficiency or immunosuppressive treatment (6, 9.8% and 5, 8.2%), respectively. Species identification within the complex was possible in 68.9% (42/61) of cases: 32 (76.2%) of *C. haemulonii*, 17 (40.4%) of *C. duobushaemulonii*, and 7 (16.6%) harboring both species. Of note, *C. parapsilosis* was concomitantly identified in 27 (44.3%) of these samples. Three patients presented 2 positive samples more than 1 month apart, but we could not assert that they were the same species.

Patients with ophthalmologic colonization were mostly women (7/12, 58.3%); the mean age was 35 years; all lived in Martinique and presented no comorbidity, except wearing contact lenses (100%). All had keratitis or corneal abscess due to *Pseudomonas aeruginosa* or *Fusarium solani* complex.

The 17 subjects of the ICU group (**Table 2**) were mostly men (13/17, 76.5%), with a mean age of 62.3 years, living exclusively in the West Indies. The main comorbidities were diabetes mellitus (4, 23.5%), surgery in the previous month (4, 23.5%), and recent exposure to antifungals (6, 35.3%). The median length of hospitalization was 15 days. The most represented specimens were isolated from the respiratory tract (12, 70.6%), including 10 (83.3%) performed on

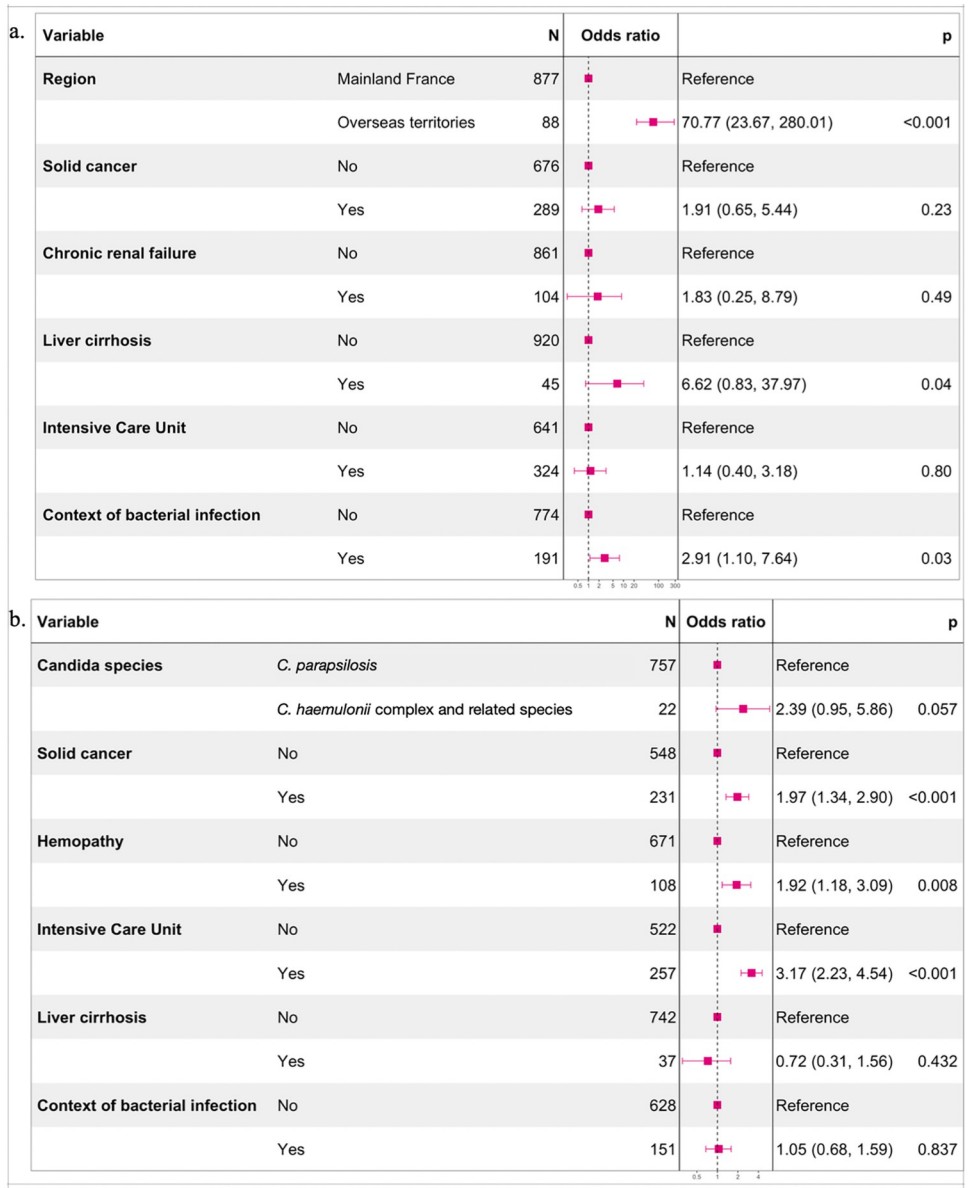

**Fig 2. Explanatory models for the occurrence and the mortality of the fungemia.** a. Explanatory model for the occurrence of *C. haemulonii complex*, *C. pseudohaemulonii* or *C. vulturna* fungemia rather than *C. parapsilosis* (adjusted odds ratio from logistic regression coefficients with 95% confidence interval). b. Explanatory model for the death within 3 months of *C. haemulonii complex*, *C. pseudohaemulonii*, *C. vulturna* or *C. parapsilosis* fungemia (adjusted odds ratio from logistic regression coefficients with 95% confidence interval).

intubated patients. *C. haemulonii* and *C. duobushaemulonii* were equally frequent, and one patient had mixed colonization. Notably, *C. parapsilosis* was also identified in 5 (29.4%) of these samples. Among the 12 patients screened for multi-drug resistant bacteria, 5 (41,6%) were positive. The crude mortality at 3 months was 47.1%.

## Literature review

The search yielded 274 fungemia from 14 countries (**Fig 3**), of which 218 were excluded, mainly due to the lack of individual data (**S3 Fig**). Finally, 56 cases were selected; all occurred

**Table 2. Characteristics of patients with *C. haemulonii* complex colonization in university hospital of Martinique between 2014 and 2020.**

| Characteristics | Outpatients[a] | Patients in ICU[b] |
|---|---|---|
| Total of events | 87 | 17 |
| Mean age in years (SD) | 57.8 (16.7) | 62.2 (17.6) |
| Male (%) | 34 (39.1) | 13 (76.5) |
| Birth continent (%) | | |
| America | 81 (93.1) | 15 (88.2) |
| Europe | 6 (6.9) | 2 (11.8) |
| Residence department (%) | | |
| Côtes-d'Armor (Mainland France) | 1 (1.1) | 0 (0.0) |
| Guadeloupe (FWI) | 0 (0.0) | 2 (11.8) |
| Martinique (FWI) | 86 (98.9) | 15 (88.2) |
| Profession (%) | | |
| Catering | 3/77 (3.9) | 0/15 (0.0) |
| Construction | 5/77 (6.5) | 2/15 (13.3) |
| Education | 5/77 (6.5) | 0/15 (0.0) |
| Farmer | 1/77 (1.3) | 1/15 (6.7) |
| Gardener | 0 (0.0) | 1/15 (6.7) |
| Healthcare | 14 (18.2) | 0/15 (0.0) |
| Retired | 24/77 (31.2) | 2/15 (13.3) |
| Unemployed | 12/77 (15.6) | 3/15 (20.0) |
| Other | 13/77 (16.8) | 2/15 (13.3) |
| Travels (%) | 1/57 (1.8) | 0/13 (0.0) |
| Aquatic activity (%) | 2/21 (9.5) | 0/2 (0.0) |
| Gardening (%) | 6/21 (28.6) | 1/2 (50.0) |
| Mean BMI in Kg/m$^2$ (SD) | 26.79 (7.6) | 27.4 (8.2) |
| Solid cancer (%) | 14 (16.1) | 1 (5.9) |
| Hematological malignancy (%) | 4 (4.6) | 1 (5.9) |
| Solid organ transplant (%) | 2 (2.3) | 0 (0.0) |
| Diabetes mellitus (%) | 21 (24.1) | 4 (23.5) |
| Corticosteroids (%) | 3 (3.4) | 2 (11.8) |
| Immunosuppressive treatment (%) | 9 (10.3) | 0 (0.0) |
| HIV (%) | 1 (1.1) | 1 (5.9) |
| Other immunodepression (%) | 9 (10.3) | 1 (5.9) |
| Surgery during the previous month (%) | 1 (1.1) | 4 (23.5) |
| Exposure to antifungal agent in the previous month (%) | 3 (3.4) | 6 (35.3) |
| Median length of stay in hospital when sampling in days [IQR] | 0.2 (0.5) | 38.1 (35.7) |
| Sampling site (%) | | |
| Ear, Nose, or Throat | 4 (4.6) | 1 (5.9) |
| Genital | 1 (1.1) | 0 (0.0) |
| Ophthalmologic | 12 (13.8) | 0 (0.0) |
| Respiratory tract | 9 (10.3) | 12 (70.6) |
| Pus[c] | 0 (0.0) | 2 (11.8) |
| Skin | 61 (70.1) | 1 (5.9) |
| Urine | 0 (0.0) | 1 (5.9) |
| Species (%)[d] | | |
| *C. duobushaemulonii* | 14 (16.1) | 7 (41.2) |
| *C. haemulonii sensu lato* | 33 (37.9) | 7 (41.2) |
| *C. haemulonii sensu lato* + *C. duobushaemulonii* | 7 (8.0) | 1 (5.9) |

*(Continued)*

**Table 2.** (*Continued*)

| Characteristics | Outpatients[a] | Patients in ICU[b] |
|---|---|---|
| *C. haemulonii* complex | 33 (37.9) | 2 (11.8) |

SD: Standard deviation; FWI: French West Indies; IQR: Interquartile Range

[a] Samples collected during ambulatory care (<48 hours of hospitalization)

[b] Samples collected in ICU at least 48 hours after admission

[c] Superficial swabbing, considered as skin contaminant

[d] Identification accuracy depending on the method used (biochemical, MALDI-TOF MS)

between 2006 and 2022. Cases originated mainly from tropical regions (44/56, 78,6%) (**S2 Table**). We individualized two age groups: children (n = 19) with a median age of 3.5 years (IQR = 0.1–5), and adults (n = 13), with a median age of 66 years (IQR = 52–82). There was no gender predominance in adults. As in our fungemia series, *Candida haemulonii sensu stricto* was the dominant species. Among adults, the distribution of species varied according to age, with a median age of 79 years for *C. haemulonii sensu stricto* compared to 56, 49, and 59 years for *C. duobushaemulonii*, *C. haemulonii var. vulnera*, and *C. pseudohaemulonii*, respectively. Age was known for only one case of *C. vulturna* (83 years).

Although previous antifungal exposure was more frequent in the literature than in our series, the susceptibility profile was similar, with high MIC values for amphotericin B and fluconazole (**S4 Fig**). The median MIC values of amphotericin B were high, notably for *C. pseudohaemulonii* (32 mg/l) *C. vulturna* (8 mg/l), and *C. duobushaemulonii* (4 mg/l). The median MIC values of fluconazole were high, especially for the *C. haemulonii* complex yeasts (64 mg/l). Micafungin MIC values were 4 mg/l for 3 strains and ≤0.500 mg/l for all the others.

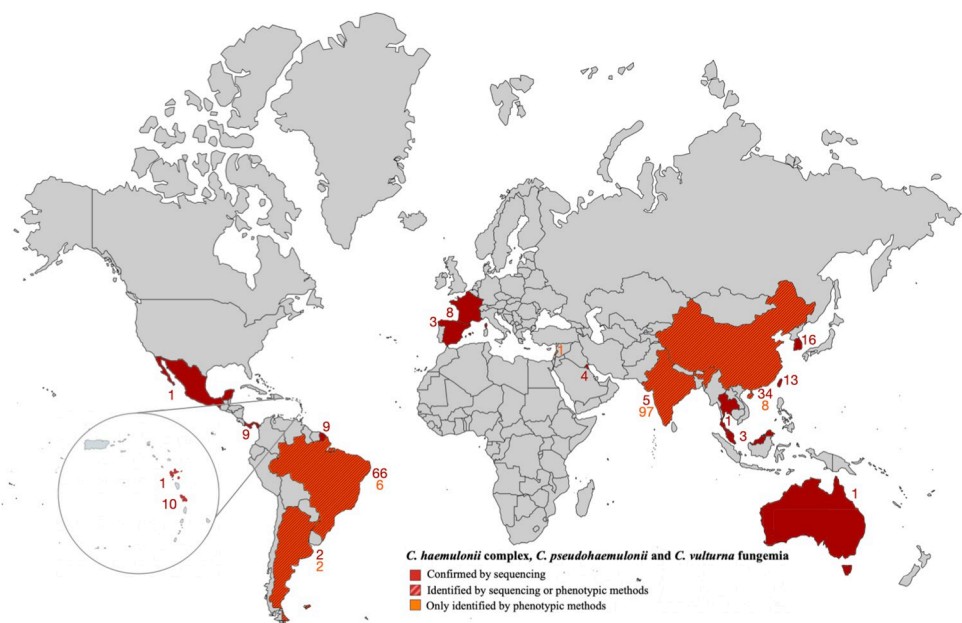

**Fig 3. World mapping of cases of fungemia due to *C. haemulonii* complex, *C. pseudohaemulonii* or *C. vulturna*.** According to our case series (obtained from the YEASTS program 2002–2021 and the RESSIF Network 2012–2021) and the literature review (Medline, 1962–2022). References are with the S3 Fig. Map created from fla-shop.com (https://www.fla-shop.com/svg/, CC BY 4.0 license) modified with inkscape software.

Among all cases, the crude mortality rate was high (5/12, 41,7%) and comparable to our fungemia series, although the time to death was not specified in most articles.

## Discussion

We report a large series of fungemia and colonization caused by emerging multiple resistant yeasts of the *C. haemulonii* complex, or very closely related species (*C. pseudohaemulonii* and *C. vulturna*), and a case-control study with *C. parapsilosis* as reference. We thereby report the first cases of *C. vulturna* fungemia in South America and Europe since its description in South-East Asia in 2016 [11]. It is also the most important literature review with individual data of fungemia caused by these yeasts and molecularly proven to avoid misidentification.

Considering our series, these rare fungemia are particularly present in tropical regions. Indeed, among centers of the RESSIF and/or the YEASTS programs, only 9 reported *C. haemulonii* complex-related species fungemia; of these, 3 declarative centers were located in tropical regions, accounting for 20/28 (71.4%) of the whole series. Living in the tropics is by far the factor most associated with the occurrence of fungemia due to these yeasts rather than with *C. parapsilosis.*

In our series, 92.9% patients presented risk factors comparable to those of *C. parapsilosis* fungemia, which also corresponds to that observed elsewhere [27]. Contrastingly, pre-exposure to azoles was uncommon in our series (16.6%), although it is usually considered a significant contributor to non-*albicans* candidemia [28,29]. The high rate of central venous catheters (68.2%) and the absence of digestive surgical site infections suggest skin as the source for *C. haemulonii* complex-related species candidemia. This is also supported by our results on skin colonization and the ability of such pathogens to form biofilms on prosthetic materials [6,7].

Fungemia caused by *C. haemulonii* complex yeasts and related species seems to occur in patients at risk of skin-related fungemia, mainly in tropical regions. Case-control study with *C. parapsilosis* fungemia and the study of carriage, both in the community and in ICU in Martinique allow us to assume that these fungemia are frequent in tropical regions not because of specific host characteristics, but rather because of the burden causing by these yeasts in the environment, which favors carriage both in the outpatients and inpatients.

The broad antifungal resistance of *C. haemulonii* complex-related species makes them potentially difficult to treat. Although there are no clinical breakpoints or ECOFFS, the particularly high MIC values for amphotericin B, fluconazole, and to a lesser extent other azoles, suggest intrinsic resistance to these antifungals. Correct identification of *C. haemulonii* complex-related species is relevant not only for epidemiological surveillance but has also a direct therapeutic impact, with amphotericin B having a particularly low *in vitro* activity against *C. duobushaemulonii* and *C. pseudohaemulonii* isolates for instance [30]. Unlike some previously published studies [10,15,16], no death appeared to be attributable to treatment failure in our series, even for the 3 patients treated with fluconazole or itraconazole (whose strains had exceptionally low MIC values for fluconazole ($\leq$16 mg/L).

All-cause mortality was similar between our series (44%) and the literature review (46%). As already shown for catheter-related candidemia, rapid control of the cutaneous portal of entry may play a critical role in preventing death [31,32].

In Martinique, and probably also in Latin America and other Caribbean territories, cutaneous colonization by *C. haemulonii* exists, especially outside the hospital setting. This is in contrast with its sibling species *C. auris*, which is often presented as a pure nosocomial yeast [3]. The three patients with positive skin specimens sampled more than one month apart allow us to suspect chronic colonization. In our study, apart from fungemia, no deep-seated infections were observed. Moreover, the fact that all patients with positive ophthalmologic samples wore

contact lenses and that 10 of the 12 patients with respiratory colonization in ICU were intubated, suggests colonization of foreign devices [3].

Our study has several limitations. Missing data, especially on the oldest cases, affected our results. We were notably unable to retrieve the time-to-positivity of blood cultures, the confirmation of systematic catheter removal, and the time required for sterilization of blood cultures. Moreover, no skin fungal mapping was reported, precluding confirmation of the skin colonization hypothesis.

Nevertheless, according to our study, the presence of yeasts in a blood culture of a patient hospitalized or coming from a tropical region, especially if presenting a central veinous catheter, should suggest the diagnosis of *C. haemulonii* complex-related yeasts fungemia, as for *Candida parapsilosis*. Such a situation requires management of the infection's portal of entry and reinforces the utility of probabilistic treatment with echinocandins. Despite the lack of specific clinical trials, there are some *in vitro* and animal data [33] as well as case reports of treatment failures with azoles or amphotericin B [10,15] that justify the use of echinocandins as first-line therapy. When rapid identification by MALDI-TOF MS is not available, the particular aspect of the cultures on chromogenic media might be sufficient to support the diagnosis [34]. This strategy could limit the delay before initiation of appropriate treatment, which is strongly correlated with mortality in candidemia [31,35]. Horizontal transmission capacity, especially in hospitals, has already been suggested by outbreaks involving the *C. haemulonii* complex [15,36]. The emergence of these yeasts is also supported at the genomic level: the characteristics shared by *C. auris* and the *C. haemulonii* complex-related species (synteny, similar gene family expansions) suggest similarities in the ecology and physiology of these species and the potential for them to also emerge as dangerous pathogens [37].

Historically described as warm sea yeasts [38], the *C. haemulonii* complex-related yeasts were later identified as saprophytes of tropical aquatic and terrestrial environment [39–41]—for instance, *C. vulturna* has been isolated from a tropical flower in the Philippines [11]—and as a potential pathogen for some animal species [9,42]. Some countries have recently reported cases in neonatology, suggesting the presence of these yeasts in artificial environments. Although we have no evidence in our series for an epidemic event, other authors have reported a series of clustered cases [15,30], with the same clone of *C. pseudohaemulonii*, raising the possibility of nosocomial transmission. A scenario of an emerging epidemic clone within these multi-drug resistant yeast populations is therefore quite credible [30], like the *C. auris* healthcare-associated outbreaks described since the 2010s.

## Conclusion

Fungemia caused by *C. haemulonii* complex and related species yeasts are rarely observed in France and occur mainly in overseas tropical territories. These environmental yeasts colonize the skin of general population, as well as medical devices, favoring skin-related fungemia in patients with risk factors. These multi-drug resistant yeasts share numerous characteristics with *C. auris*, and deserve to be known and monitored.

## Supporting information

**S1 Fig. Bar plot representing the temporal distribution of *C. haemulonii* complex, *C. pseudohaemulonii* and *C. vulturna* fungemia (YEASTS and RESSIF databases).** (PDF)

**S2 Fig. Flow chart of the samples with positive culture for *C. haemulonii* complex yeasts considered nonpathogenic, isolated in Martinique University Hospital (French West**

**Indies) between 2014 and 2020.**
(PDF)

**S3 Fig. Flow chart of the reviewed cases of *C. haemulonii* complex, *C. pseudohaemulonii* and *C. vulturna* fungemia from MedLine (1962–2021).**
(PDF)

**S4 Fig. Susceptibility profiles of strains from the literature review (MedLine 1962–2022), by microdilution (n = 36) or commercial methods (n = 16), or not specified (n = 1).**
(PDF)

**S1 Table. Confirmation of identification by MALDI-TOF MS for strains initially identified *C. haemulonii* complex by API ID32C (BioMerieux, Marcy l'Étoile, France) and stored at the University hospital of Martinique.**
(PDF)

**S2 Table. Characteristics of patients and strains regarding fungemia found in the literature review with individual data of *C. haemulonii* complex or related yeasts fungemia identified by molecular sequencing (MedLine 1962–2022).**
(PDF)

**S1 Acknowledgments.** Members of the French Mycoses Study group who contributed to the data are in alphabetical order of the cities.
(PDF)

## Author Contributions

**Conceptualization:** Ugo Françoise, Marie Desnos-Ollivier, Nicole Desbois-Nogard, Olivier Lortholary.

**Data curation:** Marie Desnos-Ollivier, Yohann Le Govic, Karine Sitbon, Ruddy Valentino, Sandrine Peugny, Taieb Chouaki, Edith Mazars, André Paugam, Muriel Nicolas, Nicole Desbois-Nogard.

**Formal analysis:** Ugo Françoise.

**Methodology:** Ugo Françoise, Marie Desnos-Ollivier, Nicole Desbois-Nogard, Olivier Lortholary.

**Resources:** Yohann Le Govic.

**Supervision:** Marie Desnos-Ollivier, Nicole Desbois-Nogard, Olivier Lortholary.

**Writing – original draft:** Ugo Françoise.

**Writing – review & editing:** Marie Desnos-Ollivier, Yohann Le Govic, Karine Sitbon, Ruddy Valentino, Sandrine Peugny, Taieb Chouaki, Edith Mazars, André Paugam, Muriel Nicolas, Nicole Desbois-Nogard, Olivier Lortholary.

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
