## [Decision Letter · Decision Letter 0]

8 Apr 2023

Dear Dr Lortholary,

Thank you very much for submitting your manuscript "Candida haemulonii complex, an emerging threat from tropical regions?" for consideration at PLOS Neglected Tropical Diseases. As with all papers reviewed by the journal, your manuscript was reviewed by members of the editorial board and by several independent reviewers. The reviewers appreciated the attention to an important topic. Based on the reviews, we are likely to accept this manuscript for publication, providing that you modify the manuscript according to the review recommendations. 

Sincerely,

Joshua Nosanchuk, MD

Section Editor

Joshua Nosanchuk

Section Editor

Reviewer's Responses to Questions

**Key Review Criteria Required for Acceptance?**

**Methods**

-Are the objectives of the study clearly articulated with a clear testable hypothesis stated?

-Is the study design appropriate to address the stated objectives?

-Is the population clearly described and appropriate for the hypothesis being tested?

-Is the sample size sufficient to ensure adequate power to address the hypothesis being tested?

-Were correct statistical analysis used to support conclusions?

-Are there concerns about ethical or regulatory requirements being met?

Reviewer #1: Objectives of the study are clearly stated. Study design is appropriate for the stated objectives. Population described clearly and appropriate for objectives of study. Sample size is small but this is a rare disease so I think the author have done well in combining data from multiple years as well as performing a literature review. This has added weight to the findings and helped overcome the challenge of small numbers. No ethical concern.

Reviewer #2: The objective of the work is clear, with this research it was expected to know a little more about the C. haemulonii complex and the species that are included in it. It is a work of many years and that implied a complete experimental design, which is reflected here. All methodologies were adequate to meet the objective of the work. It should be noted that to contrast the results, epidemiological data from the region were used, which were obtained in different hospitals and foundations, in addition, these data were compared with what has already been reported in the literature. There were many samples of patients analyzed, therefore, the population n is sufficient for the statistical analyzes to be more complete. With respect to bioethical permits, initially they mention the institutions by which the work was approved, however, nothing is mentioned about the informed consent of the patients, or if this was not required.

Reviewer #3: -Are the objectives of the study clearly articulated with a clear testable hypothesis stated? Yes.

-Is the study design appropriate to address the stated objectives? Yes.

-Is the population clearly described and appropriate for the hypothesis being tested? Yes.

-Is the sample size sufficient to ensure adequate power to address the hypothesis being tested? Yes, though samples sized were relative small.

-Were correct statistical analysis used to support conclusions? Yes.

-Are there concerns about ethical or regulatory requirements being met? No.

**Results**

-Does the analysis presented match the analysis plan?

-Are the results clearly and completely presented?

-Are the figures (Tables, Images) of sufficient quality for clarity?

Reviewer #1: Analysis matches the presented plan. Results are very clear and no issues with the tables.

- Please highlight the major differences in the two groups from Table 1.

- Line 198 - Is the present of that yeast in the eye only a colonizer? Could it have a role to play in the patient's keratitis/corneal abscess?

Reviewer #2: The data analysis is understandable, the tables are a good complement to understand what was found. As has been previously reported in other studies, the incidence of candidiasis increases especially in immunocompetent and immunosuppressed patients, an example of this is what was found in this study. Cancer and diabetes mellitus are diseases that are associated with an increase in candidiasis caused by these species of the complex. All the findings are well explained and are supported by tables and figures. I suggest that the figures and tables found in supplementary material be revised and modified, since they are of poor quality and much information is lost.

Reviewer #3: -Does the analysis presented match the analysis plan? Yes.

-Are the results clearly and completely presented? Yes.

-Are the figures (Tables, Images) of sufficient quality for clarity? Yes.

**Conclusions**

-Are the conclusions supported by the data presented?

-Are the limitations of analysis clearly described?

-Do the authors discuss how these data can be helpful to advance our understanding of the topic under study?

-Is public health relevance addressed?

Reviewer #1: The conclusions are supported by the presented data. Limitations of the study are clearly described. This is helpful from a public health standpoint and a significant contribution to the understanding of this rare group of yeast that used to be misidentified as C. auris by some identification systems.

Reviewer #2: With respect to the conclusions section, I consider that these should be mentioned in the final part of the text. In the discussion section some limitations of the work are mentioned and how they could be solved, however, in the literature there is a lot of information that could help to complement this section. In the attached file there are some comments on this specific section. At the end of the paper it is necessary to mention the future perspectives of this work, why it is important to study these species, especially in the hospital environment, and the implications it has on public health.

Reviewer #3: -Are the conclusions supported by the data presented? Yes.

-Are the limitations of analysis clearly described? Yes.

-Do the authors discuss how these data can be helpful to advance our understanding of the topic under study? Yes.

-Is public health relevance addressed? Yes.

**Editorial and Data Presentation Modifications?**

Reviewer #1: Minor revisions as suggested in the textboxes above and below.

Reviewer #2: Throughout the following text I mention some points that should be taken into account for this revision. The changes are minor. Special attention should be paid to the spelling of the reference section.

Reviewer #3: (No Response)

**Summary and General Comments**

Reviewer #1: - First page - Abstract/Results: a psace is missing "fungemiadue" and "werefrequently"

- Long form of ICU needs to be provided before it can be used as an acronym (line 107)

- I would mention that this specie could be a C. auris misidentification on identification by Vitek 2 version 8.01

- I suggest adding the following reference to line 255-256 : Microorganisms. 2020 Feb; 8(2): 181.

Published online 2020 Jan 28. doi: 10.3390/microorganisms8020181 PMCID: PMC7074697 PMID: 32012865

- I suggest reviewing "effraction" on line 267 as I am familiar with it as a francophone but this term is not commonly used in the English language to mean what its intended meaning in French

Reviewer #2: General comments are available in the following text. I consider it to be an interesting and novel text, as there are few works that deal with all species of the C. haemulonii complex. In addition, it is the first report of these species in France, and to achieve this it was necessary to review biological samples from 2002 to 2021, which indicates that it is a very complete report. Contributing to the knowledge of these species has important implications for science, thanks to these studies it is possible to search for new alternative treatments for the control of these species. In addition, this information provides interesting data about the biology of the species of this complex, from their lifestyle in the environment to how they infect their host. It would be interesting to emphasize the identification of these species, since it is known that distinguishing between species is still a complicated task and in some cases unsuccessful.

Reviewer #3: General comments

1. I review this manuscript entitled “Candida haemulonii complex, an emerging threat from tropical regions?” with interest. This manuscript described clinical and demographic characteristics of patients with fungemia due to C. haemulonii complex and related species (C. pseudohaemulonii, C. vulturna). 

2. This is a neglected tropical disease warrants more attention because C. haemulonii complex (C. haemulonii sensu stricto, C. duobushaemulonii, C. haemulonii var. vulnera) and related species (C. pseudohaemulonii and C. vulturna), are phylogenetically close to Candida auris with intrinsic antifungal resistance. I fully agree what authors emphasized “Multidrug-resistant C. haemulonii complex-related species are responsible for fungemia and colonization in community and hospital settings, especially in tropical regions, warranting closer epidemiological surveillance to prevent a potential C. auris-like threat.” (Abstract, Conclusions/Significance). However, its importance is underestimated due to difficulty in accurate identification and thus, underreported. 

3. Due to limited case numbers and published data, this manuscript included three part of results. A nationwide case series reported in France during 2002-2021 (28 cases, the first epidemiological data on these yeasts in France), a single center study regarding colonization and infection due to C. haemulonii complex (40 cases), and a literature review of fungemia due to C. haemulonii complex and related species reported in Medline (1962-2022) (274 cases). The final part included all reported cases of C. haemulonii complex, C. pseudohaemulonii, and C. vulturna fungemia identified by sequencing, for which individual clinical information and/or susceptibility to antifungals were available.

Specific comments

Lines 158-160, please shorten as these data were presented in the Table 1. 

Lines 161-163, “Four distinct species were identified….”, thus, suggest delete the same content in Table 1.

Table 1. Among patients with available birth continent 18 of 20 (90.0%) patients with C. haemulonii complex-related species were America. Is it due to the majority of cases were reported from oversea France?

PLOS authors have the option to publish the peer review history of their article (what does this mean?). If published, this will include your full peer review and any attached files.

Reviewer #1: Yes: Samuel Bourassa-Blanchette

Reviewer #2: Yes: Manuela Gómez Gaviria

Reviewer #3: No

Figure Files:

Data Requirements:

Reproducibility:

References

---

## [Editor Report · Decision Letter 1]

9 Jun 2023

Dear Dr Lortholary,

We are pleased to inform you that your manuscript 'Candida haemulonii complex, an emerging threat from tropical regions?' has been provisionally accepted for publication in PLOS Neglected Tropical Diseases. Thank you for providing a thoughtful and comprehensive response to the reviewer comments! 

Best regards,

Joshua Nosanchuk, MD

Section Editor

---

## [Editor Report · Acceptance letter]

25 Jul 2023

Dear Dr Lortholary,

We are delighted to inform you that your manuscript, " *Candida haemulonii* complex, an emerging threat from tropical regions?," has been formally accepted for publication in PLOS Neglected Tropical Diseases.

Best regards,

Shaden Kamhawi

co-Editor-in-Chief

Paul Brindley

co-Editor-in-Chief
